# Reforming the Mechanism: Editing Reasoning Patterns in LLMs with Circuit Reshaping

**Zhenyu Lei**[1,*] **Qiong Wu**[2]**, Jianxiong Dong**[2],
**Yinhan He**[1]**, Emily Dodwell**[2]**, Yushun Dong**[3] **& Jundong Li**[1]
[1]University of Virginia, [2]AT&T Chief Data Office, [3]Florida State University
`{vjd5zr, nee7ne, jundong}@virginia.edu,`
`{qw6547, JD612R, ed720d}@att.com, yushun.dong@fsu.edu`

## Abstract

Large language models (LLMs) often exhibit flawed reasoning ability that undermines reliability. Existing approaches to improving reasoning typically treat it as a general and monolithic skill, applying broad training which is inefficient and unable to target specific reasoning errors. We introduce *Reasoning Editing*, a paradigm for selectively modifying specific reasoning patterns in LLMs while preserving other reasoning pathways. This task presents a fundamental trade-off between *Generality*, the ability of an edit to generalize across different tasks sharing the same reasoning pattern, and *Locality*, the ability to preserve other reasoning capabilities. Through systematic investigation, we uncover the *Circuit-Interference Law*: Edit interference between reasoning patterns is proportional to the overlap of their neural circuits. Guided by this principle, we propose *REdit*, the first framework to actively reshape neural circuits before editing, thereby modulating interference between reasoning patterns and mitigating the trade-off. REdit integrates three components: (i) *Contrastive Circuit Reshaping*, which directly addresses the generality-locality trade-off by disentangling overlapping circuits; (ii) *Meta-Contrastive Learning*, which extends transferability to novel reasoning patterns; and (iii) *Dual-Level Protection*, which preserves preexisting abilities by constraining reshaping update directions and regularizing task-level predictions. Extensive experiments with `Qwen-2.5-3B` on propositional logic reasoning tasks across three difficulty levels demonstrate that REdit consistently achieves superior generality and locality compared to baselines, with additional validation in mathematics showing broader potential. Our code is available at https://github.com/LzyFischer/REdit.

## 1 Introduction

Large language models (LLMs) achieve state-of-the-art performance across various domains such as mathematics (Liu et al., 2023a; 2024a), law (Cheong et al., 2024; Sun, 2023), and medicine (Zhao et al., 2023; Hadi et al., 2023). The success arises from their exceptional reasoning ability when executing complex instructions (Lu et al., 2023; Villalobos et al., 2022; Wang et al., 2024b). Despite this success, LLMs often produce incorrect or misleading responses (Perković et al., 2024; Huang et al., 2025) driven by spurious reasoning processes, which significantly undermines their reliability and safety. For example, an LLM may correctly encode the fact that "`if a brain aneurysm is present, a CT scan will show bleeding or swelling (A→B OR A→C)`", but still wrongly infer "`no bleeding implies no aneurysm (¬B → ¬A)`", risking harmful medical consequences (Sim & Chen, 2024). Addressing such gaps remains a critical challenge for researchers and practitioners alike.

To strengthen reasoning, researchers typically view it as one general, monolithic skill that calls for broad enhancement (Wang et al., 2023; Parmar et al., 2024; Wan et al., 2024). Standard approaches include fine-tuning on large reasoning corpora (Zhang et al., 2024a; Kumar et al., 2025),

---

*This work was initiated and completed while Zhenyu was an intern with AT&T CDO. Both Qiong Wu and Jundong Li are the corresponding authors.

reinforcement learning from human feedback (RLHF) (Havrilla et al., 2024a; Yue et al., 2025), and sophisticated test-time prompting (Bi et al., 2024; Zhang et al., 2022). However, treating the LLM's reasoning as a monolithic ability has several drawbacks. First, overall reasoning enhancement can be difficult and expensive, demanding extensive human annotation and huge computational budgets (Luo et al., 2024; Lai et al., 2025). Second, growing evidence indicates that LLMs' reasoning is not monolithic but can be decomposed into separable patterns (Zhang et al., 2025b; Jiang et al., 2025; Zhang et al., 2025d; Shao & Cheng, 2025). Indiscriminately training over every reasoning pattern fails to distinguish between those the model already handles well and those it struggles with, thus leading to inefficient use of resources and suboptimal correction of specific reasoning errors. Therefore, recent approaches have shifted towards enhancement at the level of specific reasoning trajectories or intermediate steps, which involve only a handful of reasoning patterns (Cui et al., 2025; Havrilla et al., 2024b). However, these methods heavily rely on the model's own self-verification often without the model truly mastering the correct reasoning patterns, thus failing to reliably remedy reasoning errors. As a result, how to correct erroneous and inject new reasoning patterns without retraining on the whole reasoning datasets still remains an open problem. Recent work has demonstrated that specific reasoning patterns are encoded in localized parameters or neural circuits within LLMs (Hong et al., 2024; Kim et al., 2024), mirroring the way factual knowledge is stored in model weights (Meng et al., 2022a; Yao et al., 2024; Zhang et al., 2024c). Given the success of parameter-based methods for editing piecewise knowledge in LLMs (De Cao et al., 2021; Meng et al., 2022b), we propose a natural extension: ***If knowledge can be edited through parameter modification, can we analogously edit LLMs to correct flawed reasoning patterns or inject new ones?***

In this paper, we take an initial step toward reasoning editing, defined as the selective modification of a certain LLM's reasoning pattern while preserving its factual knowledge and other reasoning pathways. To establish a rigorous foundation for this investigation, we focus on propositional logic (PL), where reasoning patterns can be precisely defined and systematically evaluated. Although structurally simple, reasoning editing in PL remains challenging due to two fundamental desiderata (Hua et al., 2024; Sun, 2025): *(1) Generality*, edits applied to one instance should consistently generalize to all instances with the same reasoning pattern across domains, rather than memorizing surface semantics. For example, editing the transitive rule "A→B, B→C⇒A→C" in math should also hold in medicine. *(2) Locality*, edits must remain narrowly scoped, correcting the targeted inference rule without impairing the LLMs' performance on other reasoning patterns it already handles correctly. For example, editing the spurious rule "¬B→¬A⇒A→B" should not affect modus tollens "(A→B, ¬B) ⇒¬A".The two desiderata constitute a trade-off as shown in Section 2.2, whereby enhancing one dimension typically diminishes the other, thus presenting a significant dilemma.

To tackle this trade-off, we first probe the mechanism underlying reasoning edits. Motivated by evidence that reasoning mechanism can be faithfully revealed by neural circuits, we conduct a systematic investigation into the relationship of edit effects and the circuit of reasoning pattern. Through this analysis, we discover a fundamental principle we term the ***Circuit-Interference Law***: the degree to which an edit to one reasoning pattern affects another is directly proportional to the overlap between their respective neural circuits. Guided by this observation, we introduce ***REdit***, the first framework to actively reshape circuits prior to reasoning editing, enabling controlled modulation of interference among reasoning patterns. REdit employs three key components: At its core, (1) *Contrastive Circuit Reshaping* directly addresses the generality–locality trade-off by disentangling overlapping circuits to reduce cross-reasoning pattern interference which improves locality while consolidating pattern-specific circuits to promote within-reasoning pattern generality. Building upon this foundation, (2) *Meta-Contrastive Learning* enhances transfer to broader reasoning patterns beyond those observed during reshaping and (3) *Dual-Level Protection* safeguards preexisting reasoning abilities by constraining reshaping update directions via soft null-space projection and regularizing prediction distributions of reasoning tasks. After reshaping, widely used LoRA-based editing (Ge et al., 2024) suffices to achieve the desired generality and locality. We conduct extensive experiments on Qwen-2.5-3B across three propositional-logic difficulty levels, showing that REdit consistently enhances *generality* while reinforcing *locality*, surpassing strong baselines. Furthermore, additional evaluations in the mathematics domain highlight REdit's potential to generalize effectively to broader reasoning scenarios. ***Our contributions can be summarized as follows:***

- **Reasoning Editing Paradigm:** We introduce the first systematic framework for reasoning editing, extending model editing from knowledge correction to the selective modification of logical inference patterns, and formally identify the generality-locality trade-off.

- **Circuit Reshaping Methodology:** We pioneer active neural circuit modulation in LLMs, enabling principled and targeted modification of specific reasoning pathways through controlled modulation rather than passive circuit analysis.

- **Novel REdit Framework:** We propose a unified approach that synergistically combines contrastive circuit shaping, meta-contrastive learning, and dual-level protection to simultaneously achieve both broad generality and precise locality in reasoning editing.

- **Empirical Validation:** We demonstrate consistent improvements on propositional logic reasoning tasks across three difficulty levels, showing superior performance in generality and locality compared to existing editing methods.

## 2 PRELIMINARIES

### 2.1 PROBLEM FORMULATION

We study the problem of reasoning editing for LLMs in the context of propositional logic. Our goal is to enable precise modifications to an LLM's reasoning behavior, ensuring it adheres to desired logical rules while preserving its existing correct ones. To formalize this, we first introduce the necessary components of propositional logic reasoning, then define reasoning patterns and their neural approximations, and finally present the reasoning editing problem.

**Notations.** Let $\mathcal{X} = \{x_1, \ldots, x_m\}$ be a finite set of propositional variables (PVs), each taking a truth value in $\{\text{TRUE}, \text{FALSE}\}$. Let $\mathcal{S}$ denote a fixed set of logical connectives (e.g., $\neg, \wedge, \vee, \rightarrow$). A *premise set* $\mathcal{P}$ is a collection of well-formed formulas over $(\mathcal{X}, \mathcal{S})$ that we assume to be true. A *goal* $\mathcal{G}$ is a formula over $(\mathcal{X}, \mathcal{S})$. We use the standard entailment relation $\models$ where $\mathcal{P} \models \varphi$ means every model that satisfies $\mathcal{P}$ also satisfies $\varphi$. We write $\mathcal{Y} = \{\text{TRUE}, \text{FALSE}, \text{N/A}\}$ for the three-way status labels for $\mathcal{G}$, where N/A means "neither entailed nor refuted."

**Definition 1 (Propositional-Logic (PL) Reasoning)** *Given premises $\mathcal{P}$ and a goal $\mathcal{G}$, infer the status of $\mathcal{G}$ as (1) "TRUE" if $\mathcal{P} \models \mathcal{G}$, (2) "FALSE" if $\mathcal{P} \models \neg\mathcal{G}$, and (3) "N/A" otherwise.*

**Definition 2 (Reasoning Pattern)** *Let $\widehat{\mathcal{X}}$ be a finite set of placeholder PVs composed of symbols with no semantic meaning. A reasoning pattern is $\pi = (\mathcal{P}(\widehat{\mathcal{X}}, S), \mathcal{G}(\widehat{\mathcal{X}}, S))$, where $\mathcal{P}(\widehat{\mathcal{X}}, S)$ is a set of premises comprising placeholders and the connectives in $S$ and $\mathcal{G}(\widehat{\mathcal{X}}, S)$ is the goal.*

A *substitution* $\sigma : \widehat{\mathcal{X}} \rightarrow \mathcal{X}$ replaces each placeholder by a concrete PV, yielding the instantiated pair

$$\pi_\sigma = (\mathcal{P}_\sigma, \mathcal{G}_\sigma) = (\mathcal{P}(\sigma(\widehat{\mathcal{X}}), S), \mathcal{G}(\sigma(\widehat{\mathcal{X}}), S)),$$

where $\sigma(\widehat{\mathcal{X}})$ denotes the set of ground variables obtained by applying $\sigma$ to each placeholder. Two instances $\pi_\sigma$ and $\pi_{\sigma'}$ are said to *share the same reasoning pattern* exactly when they both derive from the same template $\pi$ under different substitutions $\sigma \neq \sigma'$. In practice, LLMs internalize reasoning rather than executing explicit symbolic logic, formalized as neural approximation.

**Definition 3 (Neural Approximation of PL)** *A parameterized language model $f_\theta$ approximates PL reasoning by mapping a concrete pair $(\mathcal{P}_\sigma, \mathcal{G}_\sigma)$ to a predicted status, as $f_\theta : (\mathcal{P}_\sigma, \mathcal{G}_\sigma) \mapsto \widehat{y} \in \mathcal{Y}$.*

**Problem 1 (Reasoning Editing)** *Suppose we have a fixed neural reasoner $f_\theta$ with parameters $\theta$, we also possess a finite revision dataset $\mathcal{D} = \{(\mathcal{P}^{(i)}, \mathcal{G}^{(i)}, \widehat{y}^{(i)}, y^{*(i)})\}_{i=1}^N$ in which each $(\mathcal{P}^{(i)}, \mathcal{G}^{(i)})$ is a concrete premise–goal pair, $\widehat{y}^{(i)} = f_\theta(\mathcal{P}^{(i)}, \mathcal{G}^{(i)})$ is the original model's prediction on that pair, and $y^{*(i)}$ is the target status we wish the edited model $f_{\theta'}$ to produce instead. Our objective of reasoning editing is to find a revised parameter vector $\theta'$ that meets below three requirements.*

*(1) Edit Success. For each sample $(\mathcal{P}^{(i)}, \mathcal{G}^{(i)}, \widehat{y}^{(i)}, y^{*(i)})$ in $\mathcal{D}$, the edited model with parameter $\theta'$ must predict exactly the desired status, shown as $f_{\theta'}(\mathcal{P}^{(i)}, \mathcal{G}^{(i)}) = y^{*(i)}$.*

*(2) Generality. Let $\pi^{(i)}$ denote the underlying reasoning pattern of the example $(\mathcal{P}^{(i)}, \mathcal{G}^{(i)})$. Once we decide to revise $f$'s behavior on one specific instantiation of $\pi^{(i)}$, we require that the edit extend*

*to all other premise–goal pairs arising from the same abstract pattern. Formally, for any substitution $\sigma$ that produces the pair $\pi_\sigma^{(i)} = (\mathcal{P}_\sigma, \mathcal{G}_\sigma)$, the edited model must satisfy $f_{\theta'}(\pi_\sigma^{(i)}) = y^{*(i)}$.*

*(3) Locality. Finally, let $\mathcal{C} = \{(\mathcal{P}, \mathcal{G}) \mid f_\theta(\mathcal{P}, \mathcal{G}) = y^*\}$ be the collection of all premise–goal pairs on which the original model's prediction $f_\theta(\mathcal{P}, \mathcal{G})$ already matches the ground truth. We demand that editing $\theta$ into $\theta'$ does not disturb any of these previously correct predictions. Equivalently, for every $(\mathcal{P}, \mathcal{G}) \in \mathcal{C}$, $f_{\theta'}(\mathcal{P}, \mathcal{G}) = f_\theta(\mathcal{P}, \mathcal{G})$.*

## 2.2 PRELIMINARY STUDY

We begin by conducting preliminary experiments on a subset of propositional logic dataset ContextHub (Hua et al., 2024), where we empirically reveal a *generality–locality* trade-off: a simple edit cannot simultaneously maximize both desiderata. Our investigation is motivated by a key observation that LLMs generally lack logical reasoning ability. As Figure 1a shows, the accuracy of LLMs answering propositional logical questions (*Reasoning*) is on average $10\%$

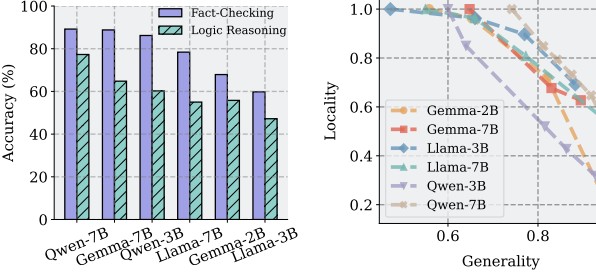

(a) Fact-Checking vs. Reasoning    (b) Generality-locality trade-off

Figure 1: LLM reasoning deficiencies and editing trade-off.

lower than tasks that merely require recalling the premise from the propositional logic (*Fact-Checking*). This gap highlights a systematic weakness in basic logical inference and motivates direct edits to correct faulty reasoning patterns.

To evaluate whether a simple edit can achieve the dual desiderata of *generality* and *locality*, we conduct experiments to measure the two metrics. Let $\Pi$ denote the index set of reasoning patterns. For each $i \in \Pi$, let $\mathcal{S}_i$ denote its instance set. Given an instance $s \in \mathcal{S}_i$, fine-tune the model on the triple $\mathcal{D}_{i,s} = (\mathcal{P}^{(s)}, \mathcal{G}^{(s)}, y^{*(s)})$ to obtain edited parameters $\theta^{(i,s)}$. The two metrics are defined as:

$$\text{Generality} = \frac{1}{\sum_i |\mathcal{S}_i|} \sum_i \sum_{s \in \mathcal{S}_i} \frac{1}{|\mathcal{S}_i \setminus \{s\}|} \sum_{(\mathcal{P}, \mathcal{G}) \in \mathcal{S}_i \setminus \{s\}} \mathbb{1}[f_{\theta^{(i,s)}}(\mathcal{P}, \mathcal{G}) = y^*(\mathcal{P}, \mathcal{G})]. \quad (1)$$

$$\text{Locality} = \frac{1}{\sum_i |\mathcal{S}_i|} \sum_i \sum_{s \in \mathcal{S}_i} \frac{1}{|\Pi \setminus \{i\}|} \sum_{j \neq i} \frac{1}{|\mathcal{S}_j|} \sum_{(\mathcal{P}, \mathcal{G}) \in \mathcal{S}_j} \mathbb{1}[f_{\theta^{(i,s)}}(\mathcal{P}, \mathcal{G}) = y^*(\mathcal{P}, \mathcal{G})]. \quad (2)$$

In practice, we approximate the last summation by randomly sampling a small subset of instances from each $\mathcal{S}_j$ instead of evaluating over the entire set for efficiency. We conduct experiments on multiple training configurations with learning rates $\eta \in [1 \times 10^{-5}, 2 \times 10^{-4}]$. As shown in Figure 1b, increasing $\eta$ improves generality but decreases locality, yielding a trade-off between generality and locality. The remainder of this work therefore proposes an framework designed to mitigate the observed trade-off, thus leading to better editing generality while preserving locality.

## 3 METHODOLOGY

### 3.1 CIRCUIT-INTERFERENCE LAW

Prior sections reveal a generality-locality trade-off: edits often fail to generalize within the intended reasoning pattern or inadvertently spill over to other ones. To understand this gap, we turn to investigate the underlying mechanisms of reasoning editing of LLMs. Recent work in mechanistic interpretability suggests that reasoning patterns are implemented by different neural circuits, and that different tasks may recruit shared modular circuits (He et al.). Building on these findings, we conjecture that the degree of overlap or separation among these circuits may govern whether edits can generalize and remain local. Intuitively, if two reasoning patterns share substantial circuit components, editing one should also influence the other; if their circuits are largely disjoint, edits are expected to remain localized. This motivates our *central hypothesis*: circuit similarity predicts cross-pattern editing effects, with closer circuits yielding stronger interference and more distant circuits preserving locality. To validate this hypothesis, we design a four-step experimental procedure.

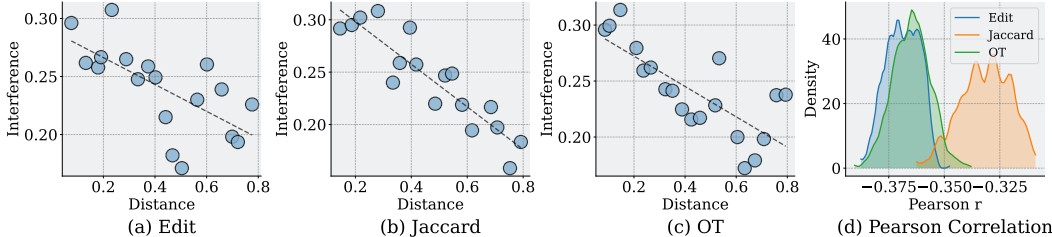

Figure 2: Correlation between circuit distance and interference. (a–c) Scatter plots with regression lines show that larger distances consistently correspond to reduced interference across different distance metrics. (d) Density plots of Pearson correlations confirm consistent negative associations.

**(1) Circuit Attribution via Edge Attribution Patching (EAP) (Syed et al., 2023).** For each pattern $\pi$, we sample $K$ instantiations $\{(\mathcal{P}_{\sigma_k}, \mathcal{G}_{\sigma_k})\}_{k=1}^{K}$ as clean input $d_k^{\text{clean}}$ and build corrupted input $d_k^{\text{patch}}$ detailed in Appendix E. Let $s_\theta(d)$ denote the log-probability of the ground-truth label $y^*(d)$. For an edge in the computational graph $e$ with activation $v_e$, its *edge attribution* for instance $k$ is an approximation of the score drop when $e$ alone is patched:

$$\text{EAP}_k(e) = \langle \nabla_{v_e} s_\theta(d_k^{\text{clean}}), v_e(d_k^{\text{patch}}) - v_e(d_k^{\text{clean}}) \rangle.$$

To mitigate instance-specific noise unrelated to the reasoning pattern, we average the edge attributions across $K$ instantiations, yielding $w_\pi(e) = -\frac{1}{K} \sum_{k=1}^{K} \text{EAP}_k(e)$. We then define the threshold $t_\pi(\tau) = \text{Quantile}_{1-\tau}(\{w_\pi(e)\})$, and construct the attributed circuit as the top–$\tau$ edges: $\mathcal{C}_\pi^{(\tau)} = \{(e, w_\pi(e)) : w_\pi(e) \geq t_\pi(\tau)\}$.

**(2) Circuit Distance.** Given two patterns $\pi_i, \pi_j$ with attributed circuits $\mathcal{C}_i^{(\tau)}, \mathcal{C}_j^{(\tau)}$, we quantify structural dissimilarity using three complementary metrics: weighted edit distance $d_{Edit}(i, j)$, Jaccard distance $d_{Jaccard}(i, j)$, and optimal transport distance $d_{OT}(i, j)$ detailed in Appendix B.

**(3) Interference from Single-Pattern Edits.** Pick a source pattern $i$ and a small revision set $\mathcal{D}_i = \{(\mathcal{P}^{(n)}, \mathcal{G}^{(n)}, y^{*(n)})\}_{n=1}^{N_i}$, where each $(\mathcal{P}^{(n)}, \mathcal{G}^{(n)})$ is an instance of $\pi_i$ and $y^{*(n)}$ its ground truth. Obtain edited parameters $\theta_{\text{edit}(i)}$ by fine-tuning $f_\theta$ on $\mathcal{D}_i$. For any target pattern $j$, define accuracy on its held-out set $\mathcal{S}_j$ as $\text{Acc}_j(\theta)$ and corresponding *edit interference* from $i$ to $j$ as $\Delta_{i \to j}$.

$$\text{Acc}_j(\theta) = \frac{1}{|\mathcal{S}_j|} \sum_{(\mathcal{P}, \mathcal{G}) \in \mathcal{S}_j} \mathbb{1}[f_\theta(\mathcal{P}, \mathcal{G}) = y^*(\mathcal{P}, \mathcal{G})], \quad \Delta_{i \to j} = |\text{Acc}_j(\theta_{\text{edit}(i)}) - \text{Acc}_j(\theta)|.$$

**(4) Circuit–Interference Relation.** We examine the correlation between interference $\Delta_{i \to j}$ and circuit distance $d(i, j) \in \{d_{\text{Jac}}, d_{\text{Edit}}, d_{\text{OT}}\}$, modeled as $\Delta_{i \to j} \approx \alpha + \beta \, d(i, j) + \epsilon$ (Figure 2a–c). We consistently find $\beta < 0$ and negative Pearson correlations, robust across edit budgets, random seeds, and dataset subsamples as illustrated in Figure 2d. We term this finding as **Circuit–Interference Law**, which posits a monotone relationship between structural proximity and cross-pattern effects where smaller circuit distance implies larger $\Delta$, and vice versa.

### 3.2 REDIT: CIRCUIT RESHAPING FOR REASONING EDITING

The **Circuit–Interference Law** suggests that achieving both generality and locality requires well-structured circuits: representations of the same reasoning pattern should align closely, while those of different patterns should remain distinct. This leads us to a bold proposition: rather than passively analyzing existing circuits, can we actively reshape them to enforce these properties? In this paper, we take a step in that direction with **REdit**, a framework that reformulates model circuits through a contrastive meta-learning objective with dual-level protection constraints before reasoning editing, enabling more effective and controlled reasoning edits.

**Contrastive Circuit Reshaping.** Directly reshaping two circuits to make them similar is challenging since (i) circuit structure is discrete and (ii) circuits are not available in closed form. We therefore adopt the *attribution weights* defined in Section 3.1 as a differentiable surrogate. Within each minibatch, we sample multiple instantiations per pattern and compute their weights $w_\pi$. We then normalize them as $\tilde{w}_\pi = w_\pi / \|w_\pi\|_2$. For each anchor example $i$, we construct a positive example $i^+$

from a different group of instantiations of the same pattern, and negatives $\mathcal{N}(i)$ from instantiations of other patterns. We then conduct InfoNCE (Oord et al., 2018) over attribution vectors:

$$\mathcal{L}_{\text{ctr}}(\theta) = -\sum_i \log \frac{\exp(\langle \tilde{w}_i, \tilde{w}_{i+} \rangle / \tau_t)}{\exp(\langle \tilde{w}_i, \tilde{w}_{i+} \rangle / \tau_t) + \sum_{j \in \mathcal{N}(i)} \exp(\langle \tilde{w}_i, \tilde{w}_j \rangle / \tau_t)} \tag{3}$$

where $\tau_t$ is temperature. Optimizing equation 3 increases similarity within a reasoning pattern and decreases similarity across patterns, shaping circuits implicitly through their attributions.

**Meta-Contrastive Learning.** Training only on observed reasoning patterns may hinder transfer to rare or unseen ones. To address this, we adopt a first-order meta-learning scheme on the contrastive objective inspired by the Meta-Contrastive Network (Lin et al., 2021), adopting a Reptile-like framework (Nichol & Schulman, 2018) that iteratively samples mini-batches, performs several inner gradient steps, and updates parameters toward task-adapted weights. By aligning gradients across tasks, this process amplifies updates along shared directions while suppressing instance-specific directions, thereby mitigating overfitting to spurious contrastive relationships between particular reasoning patterns and enabling circuits to generalize beyond those observed during training. In practice, at each meta-iteration, we sample a batch of contrastive tuples $\mathcal{B}$ each regarded as a task, perform $s$ inner steps of adaptation, and obtain task-specific parameters $\phi_i = \theta_i^s$. The outer update then moves the model weights toward the mean of these task-adapted parameters:

$$\textbf{Inner: } \theta_i^{t+1} = \theta_i^t - \alpha \nabla_\theta \mathcal{L}_{\text{ctr}}^{(i)}(\theta_i^t), \quad \theta_i^0 = \theta, \quad \textbf{Outer: } \theta \leftarrow \theta + \eta \cdot \frac{1}{|\mathcal{B}|} \sum_{i \in \mathcal{B}} (\phi_i - \theta). \tag{4}$$

**Dual-Level Protection.** To preserve the model's original behavior while enforcing correct mechanisms, we impose constraints at both the *(a) prediction level* and the *(b) optimization level*.

**(a) Prediction Distribution Preservation.** Given a correctness set $\mathcal{C}$ and a frozen reference model $f_{\theta^{\text{ref}}}$ (a pre-iteration snapshot of $\theta$), we penalize deviations on $\mathcal{C}$:

$$\mathcal{L}_{\text{pred}}(\theta) = \mathbb{E}_{(\mathcal{P},\mathcal{G}) \in \mathcal{C}} \text{KL}(f_{\theta^{\text{ref}}}(\cdot \mid \mathcal{P}, \mathcal{G}) \parallel f_\theta(\cdot \mid \mathcal{P}, \mathcal{G})). \tag{5}$$

**(b) Null-Space Protection.** At each inner step $t$ of task $i$, we form an *anchor group* $a^{(i,t)}$, with its instantiations set derived from the anchor. We compute the average prediction loss $\ell_\theta(a^{(i,t)}) = \frac{1}{|a^{(i,t)}|} \sum_{d \in a^{(i,t)}} \ell_\theta(d)$ and the gradient is $g_{i,t} = \nabla_\theta \ell_\theta(a^{(i,t)})$. To prevent reshaping from impairing reasoning task performance on the anchor, we define the rank-1 projector $\Pi_g(u) = \frac{\langle u, g \rangle}{\langle g, g \rangle + \varepsilon} g$ and the soft null-space operator $P^{(i,t)} = I - \rho \, \Pi_{g_{i,t}}$, where $\rho \in [0,1]$ controls projection strength and $\varepsilon > 0$ ensures numerical stability. The inner-loop gradients are then replaced by their projected versions:

$$\widetilde{\nabla}_\theta \mathcal{L}_{ctr}^{(i)}(\theta_i^t) = P^{(i,t)} \nabla_\theta \mathcal{L}_{ctr}^{(i)}(\theta_i^t), \qquad \theta_i^{t+1} = \theta_i^t - \alpha \widetilde{\nabla}_\theta \mathcal{L}_{ctr}^{(i)}(\theta_i^t). \tag{6}$$

When $\rho = 1$, the update is confined to the null space of $g_{i,t}$, leaving the anchor's loss unchanged to first order. While prediction preservation maintains consistency in the model's outputs, null-space protection regulates internal parameter updates, thereby preventing catastrophic drift.

**LoRA-based Edit.** After circuit reshaping, we obtain the reshaped parameters $\theta_{rsp}$. To enable fair comparison, we then apply a widely used parameter-efficient editing method LoRA on the revision set $\mathcal{D}$, yielding the adapted parameters $\theta_{\text{edit}} = \min_{\theta_{rsp}} \frac{1}{|\mathcal{D}|} \sum_{(\mathcal{P},\mathcal{G},y^*) \in \mathcal{D}} \text{CE}(f_{\theta_{rsp}}(\cdot \mid \mathcal{P}, \mathcal{G}), y^*)$, With circuit reshaping, this lightweight edit is expected to achieve improved generality and locality.

## 4 EXPERIMENTAL SETTINGS

**Datasets & Metrics.** We experiment on CONTEXTHUB (Hua et al., 2024) with details in Appendix A.1. We evaluate with the *Generality* and *Locality* metrics introduced in Section 2.2.

**Backbone LLM.** We use Qwen2.5-3B-Instruct (Yang et al., 2025) as the backbone LLM for all experiments unless otherwise noted. This model offers competitive reasoning capability at a modest parameter scale compared to larger ones, which keeps memory and inference costs manageable.

**Baselines.** We compare REdit to two families of approaches. *(i) Model Reforming:* **(1) BIMT** (Liu et al., 2023b) (Brain-Inspired Modular Training) encourages functional modularity for MLPs during pretraining; we adapt it to more complex LLMs to promote separable circuits for distinct reasoning

Table 1: Main results on ContextHub evaluated with generality and locality metrics. The best and second-best scores are highlighted in **bold** and underlined, respectively. *Raw* denotes the performance of the unedited LLM. For BIMT, we apply the same LoRA-based editing method as in REdit.

| Dataset | Metric | Raw | BIMT | LoRA | ROME | AlphaEdit | Ours |
|---------|--------|-----|------|------|------|-----------|------|
| **Level 1** | **Generality** | $60.7 \pm 2.3$ | $\underline{72.2} \pm 1.4$ | $63.8 \pm 2.9$ | $67.8 \pm 3.2$ | $67.9 \pm 1.9$ | $\mathbf{74.1} \pm 1.6$ |
|  | **Locality** | N/A | $61.5 \pm 0.7$ | $84.9 \pm 1.6$ | $\underline{89.8} \pm 3.1$ | $87.0 \pm 0.9$ | $\mathbf{94.3} \pm 0.4$ |
| **Level 2** | **Generality** | $53.2 \pm 1.4$ | $\underline{63.6} \pm 2.9$ | $58.4 \pm 0.1$ | $61.3 \pm 1.1$ | $58.8 \pm 1.5$ | $\mathbf{64.8} \pm 1.2$ |
|  | **Locality** | N/A | $59.4 \pm 4.1$ | $91.5 \pm 0.0$ | $93.1 \pm 0.1$ | $\underline{93.3} \pm 0.0$ | $\mathbf{94.3} \pm 0.5$ |
| **Level 3** | **Generality** | $45.1 \pm 1.6$ | $52.6 \pm 0.4$ | $50.1 \pm 0.8$ | $51.5 \pm 3.3$ | $\underline{54.2} \pm 0.8$ | $\mathbf{55.0} \pm 1.6$ |
|  | **Locality** | N/A | $52.3 \pm 1.0$ | $92.3 \pm 2.8$ | $\mathbf{94.6} \pm 2.7$ | $92.2 \pm 0.7$ | $\underline{94.4} \pm 0.8$ |

patterns, followed by LoRA-based editing. *(ii) Model Editing:* **(2) LoRA** (Hu et al., 2022) applies low-rank adapters for parameter-efficient fine-tuning and is a widely used and simple baseline in knowledge editing (Wang et al., 2024c; Jiang et al., 2024); **(3) AlphaEdit** (Fang et al., 2024) augments editing with null-space protection to reduce collateral changes; **(4) ROME** (Meng et al., 2022a) locates and updates internal representations associated with targeted knowledge. We adapt each method to the PL setting for a fair comparison. All editing methods select the optimal learning rate within the range $5 \times 10^{-5}$ and $2 \times 10^{-4}$. For other implementation details, refer to Appendix E.

## 5 RESULTS AND ANALYSIS

In this section, we address five research questions: **RQ1:** How does REdit compare with existing baselines? **RQ2:** What is the contribution of each component within REdit? **RQ3:** How effectively can REdit reshape circuits in LLMs? **RQ4:** To what extent does circuit reshaping transfer to unseen circuits? **RQ5:** How does REdit perform on other domains compared to baselines?

### 5.1 MAIN RESULTS

In this section, we address **RQ1** and present our findings in Table 1. Our analysis yields several key insights: (1) REdit consistently outperforms all baselines, achieving up to at most 16.1% improvements in generality and 12.2% in locality compared to LoRA without circuit shaping, and averaging 2.0% gains over state-of-the-art methods. (2) REdit's advantage increases as task complexity decreases, though improvements persist at all difficulty levels. This reflects that simpler tasks have more tractable circuit structures amenable to targeted reshaping. (3) BIMT achieves strong generality but poor locality due to its disruption of internal mechanisms, compromising preservation of original capabilities. (4) ROME and AlphaEdit exhibit compet-

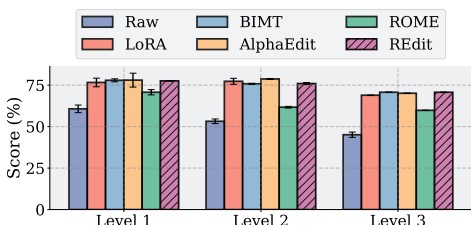

Figure 3: Editing success rates across methods on ContextHub. REdit achieves success rates comparable to other approaches, confirming that it does not compromise the model's fundamental editing capabilities.

itive locality but inferior generality. ROME's focus on middle-layer MLPs inadequately captures distributed reasoning capabilities, while AlphaEdit's constrained editing directions limit generality enhancement to preserve other knowledge.

To ensure our method does not compromise the model's fundamental editing capabilities on the target instances, we evaluate editing success rates in Figure 3. Most methods achieve comparable performance, with ROME as a notable exception showing significantly lower success rates. This result further validates that restricting modifications to middle-layer MLPs is insufficient, given that reasoning capabilities in LLMs are distributed across multiple architectural components.

### 5.2 ADDITIONAL ANALYSIS

**Ablation Study.** To address **RQ2**, we conduct an ablation study with results presented in Table 2. Here, *w/o MCL* denotes the removal of Meta-Contrastive Learning, *w/o PDP* indicates without Prediction Distribution Preservation, and *w/o NSP* represents without Null Space Protection. We have

Table 2: Ablation studies on ContextHub evaluated with generality and locality metrics. The best and second-best scores are highlighted in **bold** and underlined, respectively.

| Dataset | Metric | Raw | w/o MCL | w/o NSP | w/o PDP | **Ours** |
|---------|--------|-----|---------|---------|---------|----------|
| **Level 1** | **Generality** | $60.7 \pm 2.3$ | $72.9 \pm 0.4$ | $73.3 \pm 0.2$ | $73.4 \pm 0.5$ | **$74.1 \pm 1.6$** |
| | **Locality** | N/A | $90.7 \pm 1.8$ | $89.5 \pm 0.3$ | $90.1 \pm 2.5$ | **$94.3 \pm 0.4$** |
| **Level 2** | **Generality** | $53.2 \pm 1.4$ | $62.5 \pm 0.3$ | $62.4 \pm 1.6$ | $61.3 \pm 2.0$ | **$64.8 \pm 1.2$** |
| | **Locality** | N/A | **$94.9 \pm 0.6$** | $93.0 \pm 1.8$ | $94.0 \pm 0.8$ | $94.3 \pm 0.5$ |
| **Level 3** | **Generality** | $45.1 \pm 1.6$ | $53.8 \pm 1.3$ | $50.9 \pm 0.6$ | $51.8 \pm 0.6$ | **$55.0 \pm 1.6$** |
| | **Locality** | N/A | $93.7 \pm 1.3$ | $92.8 \pm 1.1$ | $92.8 \pm 1.2$ | **$94.4 \pm 0.8$** |

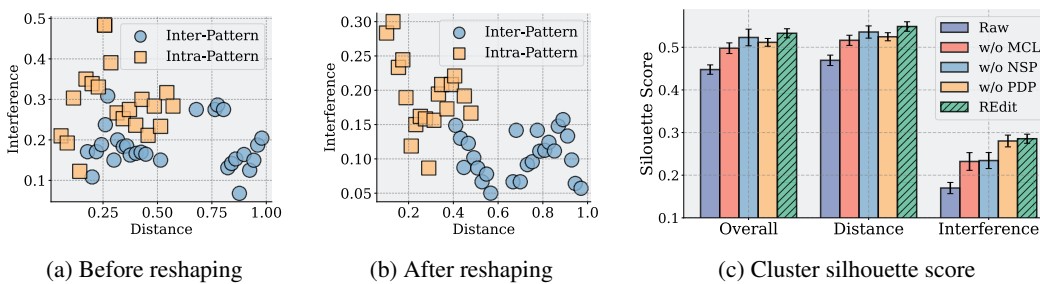

(a) Before reshaping      (b) After reshaping      (c) Cluster silhouette score

Figure 5: Circuit–interference relationship before and after circuit reshaping. (a,b) Scatter plots of intra- and inter-pattern measurements show improved separability in interference and circuit distance. (c) Silhouette scores across reasoning patterns indicate consistent gains in cluster separation.

the following observations: (1) All proposed components contribute meaningfully to REdit's overall performance, demonstrating their individual effectiveness. (2) Removing NSP or PDP substantially degrades performance, particularly in locality metrics, indicating that these protection mechanisms are essential for preserving model capabilities during circuit reshaping. (3) MCL provides modest but consistent improvements, attributable to enhanced optimization stability through meta-learning.

**Reshaping Effect on Circuit Distance.** To address **RQ3**, we measure how circuit reshaping alters circuit distances between patterns. We visualize the circuit-interference relationship as described in Section 3.1, distinguishing measurements between circuits from the same reasoning pattern (Intra-Pattern) and different reasoning patterns (Inter-Pattern). Comparing the circuit-interference relationship before and after circuit reshaping in Figure 5, we observe that the two clusters become more separable in both interference and circuit distance dimensions. The right panel shows silhouette scores for the clusters across different reasoning pattern sets, where *Overall* indicates scores in the 2-dimensional space. Our results demonstrate that REdit and its components consistently improve circuit distance separation between different reasoning patterns while refining interference patterns: increasing intra-pattern interference (enhancing generality) and decreasing inter-pattern interference (improving locality). This validates both the effectiveness of our circuit reshaping approach and the Circuit-Interference Law.

**Transferability of Reshaping.** To address **RQ4**, we investigate the transfer of the effect of meta-contrastive circuit reshaping to unseen reasoning patterns. We apply REdit to partial reasoning patterns ($20\% - 80\%$ ratio) and evaluate generality and locality on the remaining patterns. The results in Figure 4 show that while accuracy decreases slightly as the training ratio decreases, REdit consistently outperforms baselines without circuit reshaping ($0\%$ ratio) in both generality and locality metrics. This demonstrates the effectiveness of meta-contrastive learning in transferring learned circuit modifications to previously unseen reasoning patterns.

(a) Generality      (b) Locality

Figure 4: Performance on unseen reasoning patterns after circuit reshaping with different ratios for training. REdit consistently outperforms baselines without reshaping.

**Evaluation on Mathematics Tasks**   To address **RQ5**, we broaden our evaluation beyond logical tasks by assessing REdit on TemplateGSM, a mathematical reasoning benchmark. TemplateGSM encompasses multiple math templates, where each template represents a distinct reasoning pattern analogous to propositional logic reasoning patterns (detailed in Appendix A.2). The results in Figure 6 show that while all methods perform worse on TemplateGSM than on propositional logic reasoning due to the intrinsic complexity of math problems, REdit consistently outperforms all baselines, demonstrating its effectiveness on a broader range of domains. BIMT fails on both generality and locality, indicating its inability to modularize LLMs for complex tasks. Additionally, AlphaEdit and ROME show limited generality improvements, highlighting the constraints of traditional knowledge editing methods on mathematical reasoning tasks.

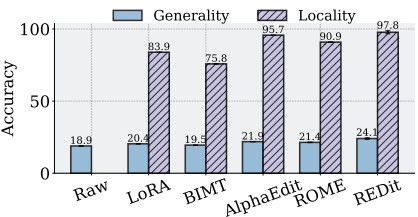

Figure 6: Evaluation on mathematical reasoning benchmark TemplateGSM.

# 6   RELATED WORKS

**LLM Reasoning.**   Recent advances in LLMs have been driven significantly by their improved reasoning ability (Huang & Chang, 2022; Yu et al., 2024; Chen et al., 2025a; Li et al., 2025; Ferrag et al., 2025; Zhang et al., 2024b; Wang et al., 2024d; Zheng et al., 2025; He et al., 2025; 2026; Zhu et al., 2025; Lei et al., 2025b), which is the capacity for structured, logical thinking to solve complex problems such as mathematical proofs (Ahn et al., 2024; Yang et al., 2024a), causal inference (Wang, 2024; Ma, 2024), and formal logic (Wan et al., 2024; Parmar et al., 2024). Despite their impressive performance, LLMs' reasoning abilities remain limited, especially with rigorous logical deduction (Cai et al., 2024), multi-hop inference (Yang et al., 2024b), and precise symbolic manipulation (Sullivan & Elsayed, 2024), thus prompting further improvement. Existing approaches often enhance reasoning through global strategies, such as supervised fine-tuning (Kumar et al., 2025; Zhang et al., 2025c; Luong et al., 2024) or RLHF (Hou et al., 2025; Yue et al., 2025; Wei et al., 2025). However, these methods treat reasoning as a monolithic capability rather than decomposing it into finer-grained, interpretable patterns (Havrilla et al., 2024b). As a result, they lack the precision to target and improve specific reasoning weaknesses (Chen et al., 2024). In this work, we propose a more granular reasoning editing paradigm that disentangles reasoning into distinct patterns. This enables targeted, efficient, and adaptive improvements tailored to specific reasoning challenges, moving beyond one-size-fits-all solutions.

**Model Editing.**   Model editing modifies a pre-trained LLM's behavior post-hoc (Wang et al., 2024c; Zhang et al., 2025a; Lei et al., 2025a), enabling error correction (Chen et al., 2025b; Li et al., 2023), knowledge updates (Wang et al., 2024a), or task adaptation without full retraining (Qi et al., 2024). Current techniques fall into several categories: memory-based methods (Liu et al., 2024b; Hu et al., 2024; Mitchell et al., 2022), meta-learning approaches (Mitchell et al., 2021; Tan et al., 2023), and localized rank-one updates (Hase et al., 2023; Meng et al., 2022a). These methods have predominantly concentrated on editing factual knowledge, typically represented as structured knowledge tuples. In contrast, reasoning editing addresses more complex reasoning processes, which are more intricately encoded within the neural circuits of LLMs (Hong et al., 2024; Kim et al., 2024). Conventional knowledge editing techniques often fail in this setting, as they struggle to satisfy the dual desiderata of generality and locality. Moreover, no prior work has systematically investigated how to directly manipulate neural circuits to enhance reasoning capabilities. In this work, we bridge this gap by taking the first step toward reasoning editing. We introduce a novel circuit-reshaping framework designed to mitigate the inherent generality–locality trade-off,thereby enabling more effective editing of reasoning patterns.

# 7   CONCLUSION

In this work, we present the first systematic study of reasoning editing, extending model editing beyond factual correction to logical inference, which introduces the generality-locality trade-off. Through circuit-level analyses, we uncover the Circuit-Interference Law, showing that interference between reasoning patterns is proportional to their circuit overlap. Inspired by this principle, we propose REdit, a framework that reshapes model circuits prior to editing to mitigate the trade-off. REdit

integrates contrastive circuit shaping to align within-pattern circuits while disentangling across-pattern ones, a meta-contrastive objective to enhance generalization, and dual-level protection to preserve both prediction distributions and update directions. Empirical results show that even with a simple LoRA editor, REdit consistently outperforms knowledge editing and model reforming baselines on propositional logic across three difficulty tiers using Qwen-2.5-3B. Additional experiments further demonstrate its potential across different reasoning domains.

## ACKNOWLEDGMENTS

Zhenyu Lei and Jundong Li are supported in part by the National Science Foundation (NSF) under grants IIS-2144209, IIS-2223769, CNS-2154962, BCS2228534, and CMMI-2411248; the Office of Naval Research (ONR) under grant N000142412636. Yushun Dong is supported by the National Science Foundation (NSF) under grants OAC-2530786 and GEO-2536578. This project was initiated while the first author, Zhenyu Lei, was a summer intern at AT&T CDO. The authors gratefully acknowledge the opportunity to participate in the internship program and appreciate the supervision and guidance provided throughout the project.

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

## ETHICS & REPRODUCIBILITY STATEMENT

We use only public, anonymized datasets. No human subjects or sensitive data are involved. The work aims to improve LLM reliability, aligning with the ICLR Code of Ethics.

For reproducibility, datasets, settings, and metrics are detailed in the paper and appendix. Code and instructions are released in the anonymous repository.

## THE USE OF LARGE LANGUAGE MODELS

In this study, we mainly leverage LLMs to enhance the clarity and polish of the manuscript. All conceptual development and methodology design were conducted by the authors.

## A   DATASET DETAILS

### A.1   PROPOSITIONAL LOGIC: CONTEXTHUB

ContextHub (Hua et al., 2024) is a benchmark for propositional logical reasoning, built on top of formal logic templates generated by DyVal (Zhu et al., 2023). It dynamically instantiated these templates into natural language questions across 11 real-world domains drawn from Wikipedia (e.g., culture, health, technology) along with an abstract form, thereby ensuring both diversity and robustness of reasoning scenarios.

**Statistics.** ContextHub consists of a total of 256 formal logic templates, spanning several difficulty levels. Each template is instantiated across 12 domains with 5 variations per domain. This yields 360 samples for level-1 logic, 600 for level-2 logic, and 2,880 for level-3 logic types. Each sample is balanced across the three answer labels (`True`, `False`, `N/A`). In this work, we treat each logic template as a distinct reasoning pattern.

**Example.** Table 3 illustrates both an abstract and a contextual instantiation of the same level-1 template. The abstract form substitutes propositional variables with arbitrary character sequences, while the contextual form grounds them in a concrete domain.

| Abstract Instance | Contextual Instance |
|---|---|
| $(\texttt{vxkgr} \lor \texttt{caunc}) \to \texttt{ybyz}$. Given `ybyz` is False, what is the value of `caunc`? | If an area of land has experienced significant uplift or been shaped by powerful erosional forces, then the terrain will feature tall, steep mountains. Given that the area does not have tall, steep mountains, can it be determined if powerful erosional forces have shaped the land? |

Table 3: Level-1 example instantiations in ContextHub.

## A.2 MATHEMATICS: TEMPLATEGSM

TemplateGSM (Zhang, 2024) is a large-scale benchmark for mathematical reasoning, constructed using the Template-based Data Generation (TDG) paradigm. Frontier LLMs (e.g., GPT-4) are employed to author parameterized meta-templates, which are then instantiated into natural language problems paired with programmatically verifiable solutions. This ensures not only linguistic and structural diversity but also guarantees correctness at scale.

**Statistics.** TemplateGSM comprises 7,473 GPT-4-authored templates, instantiated into approximately 7.47 million grade-school math problems spanning arithmetic, fractions, percentages, and elementary algebra. Problem lengths range from 18–636 tokens. In this work, we experiment on a curated subset of 600 problems, each restricted to a single numerical answer (integer or float).

**Example.** Table 4 illustrates a GPT-4-authored template alongside one instantiated problem, highlighting how TDG generates diverse mathematical reasoning tasks.

| Math Template | Instantiated Problem |
|---|---|
| [NAME] sold [NUM1] [ITEM] to [her/his/their] friends in April at a [LOCATION] in [COUNTY], [STATE]. In May, [PRONOUN] sold [NUM2] [ITEM]. How many [ITEM] did [NAME] sell altogether in April and May? | Rosy Plascencia sold 238 air fryers to her friends in April at a yoga studio boutique in Bracken County, Kentucky. In May, they sold 119 air fryers. How many air fryers did Rosy Plascencia sell altogether in April and May? |

Table 4: Example instantiations in TemplateGSM.

## B CIRCUIT DISTANCE METRIC

Given two patterns $\pi_i, \pi_j$ and their *attributed circuits* as the sets of top–$\tau$ edges ranked by attribution scores: $\mathcal{C}_\pi^{(\tau)} = \{(e, w_\pi(e)) : w_\pi(e) \geq t_\pi(\tau)\}$, we quantify structural dissimilarity between $\pi_i$ and $\pi_j$ using three complementary metrics.

**(a) Weighted Jaccard Distance (Real & Vargas, 1996).**

$$d_{\text{Jac}}(i,j) = 1 - \frac{\sum_{e \in \mathcal{C}_{\pi_i}^{(\tau)} \cup \mathcal{C}_{\pi_j}^{(\tau)}} \min\{w_i(e), w_j(e)\}}{\sum_{e \in \mathcal{C}_{\pi_i}^{(\tau)} \cup \mathcal{C}_{\pi_j}^{(\tau)}} \max\{w_i(e), w_j(e)\} + \varepsilon}. \tag{7}$$

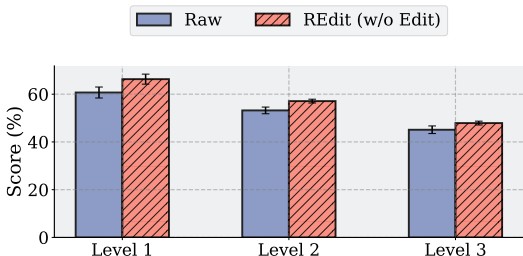

Figure 7: Comparison of accuracy between the original LLMs (*Raw*) and the unedited models after REdit circuit reshaping (*REdit (w/o Edit)*) across three difficulty levels of logical reasoning tasks. REdit consistently provides modest gains, with the most notable improvements at Level 1.

This emphasizes overlap of influential edges in the two attributed circuits.

**(b) Edit Distance (Yujian & Bo, 2007).**

$$d_{\text{Edit}}(i,j) = \frac{\sum_{e \in \mathcal{C}_{\pi_i}^{(\tau)} \cup \mathcal{C}_{\pi_j}^{(\tau)}} |w_i(e) - w_j(e)|}{\sum_{e \in \mathcal{C}_{\pi_i}^{(\tau)} \cup \mathcal{C}_{\pi_j}^{(\tau)}} \max\{w_i(e),\, w_j(e)\} + \varepsilon}. \tag{8}$$

This captures the minimal "edit cost" required to reconcile the two circuits.

**(c) Optimal-Transport (OT) Distance (Cuturi, 2013).** Normalize edge weights in each attributed circuit to probability masses

$$p_\pi(e) = \frac{w_\pi(e)}{\sum_{e' \in \mathcal{C}_\pi^{(\tau)}} w_\pi(e')}, \quad e \in \mathcal{C}_\pi^{(\tau)}.$$

Let $c(e, e') \geq 0$ denote a ground cost between edges (e.g., based on layer/head/type and token-span offsets). The optimal transport distance is then

$$d_{\text{OT}}(i,j) = \min_{T \in \Pi(p_i, p_j)} \sum_{e \in \mathcal{C}_{\pi_i}^{(\tau)}} \sum_{e' \in \mathcal{C}_{\pi_j}^{(\tau)}} T_{e,e'}\, c(e, e'),$$

$$\Pi(p_i, p_j) = \{T \geq 0 : \sum_{e'} T_{e,e'} = p_i(e),\ \sum_{e} T_{e,e'} = p_j(e')\}. \tag{9}$$

This explicitly accounts for circuit geometry by measuring the minimal mass transport needed to align the two attributed circuits.

## C  BONUS EFFECT OF REDIT

In this section, we compare the performance of the original LLMs with that of the unedited models after undergoing REdit circuit reshaping in Figure 7. Surprisingly, we observe that even without explicit editing, REdit consistently yields modest accuracy gains across three difficulty levels of logical reasoning tasks, with the largest improvements occurring on the easier problems. We attribute this phenomenon to circuit reshaping's ability to reorganize the model's internal mechanisms, where it might suppresses noisy or erroneous circuits while preserving task-critical ones, thereby enhancing the model's overall reasoning performance. We will explore this phenomenon further in the future.

## D  ALGORITHM

In this section, we provide the algorithm of REdit circuit reshaping in Algorithm 1.

---

**Algorithm 1** REdit Circuit Reshaping

---

1: **procedure** REDIT($\theta, \Pi_{\text{train}}, \eta, \alpha, s, \rho$)
2:     **Input:** LLM $\theta$; training patterns $\Pi_{\text{train}}$; rates $\eta, \alpha$; steps $s$; ratio $\rho$.
3:     **Output:** Reshaped LLM $\theta'$
4:     **Contrastive Circuit Shaping:**
5:     Derive attribution scores $\tilde{w}_\pi = w_\pi / \|w_\pi\|_2$; define InfoNCE loss $\mathcal{L}_{\text{ctr}}(\theta)$ as in Eq. equation 3
6:     **Meta-Contrastive Learning with Dual Protection:**
7:     **for** each meta-iteration **do**
8:         Sample batch $\mathcal{B} \subset \Pi_{\text{train}}$
9:         **for** each $i \in \mathcal{B}$ **do**                     ▷ Inner loop (equation 4)
10:             Initialize $\theta_i^0 \leftarrow \theta$
11:             **for** $t = 0, 1, \ldots, s - 1$ **do**
12:                 Compute preservation loss $\mathcal{L}_{\text{pred}}(\theta_i^t)$ as in Eq. equation 5
13:                 Inner objective: $\mathcal{L}_{\text{inner}}^{(i)} = \mathcal{L}_{\text{ctr}}^{(i)} + \lambda \mathcal{L}_{\text{pred}}$
14:                 Derive gradient of inner objective: $g_{i,t} \leftarrow \nabla_\theta \mathcal{L}_{\text{inner}}^{(i)}(\theta_i^t)$
15:                 Form projector $P^{(i,t)} = I - \rho \Pi_{g_{i,t}}$     ▷ Null-space protection
16:                 Update $\theta_i^{t+1} \leftarrow \theta_i^t - \alpha P^{(i,t)} g_{i,t}$     ▷ Protected update (6)
17:             **end for**
18:             Set $\phi_i \leftarrow \theta_i^s$
19:         **end for**
20:         Outer update: $\theta \leftarrow \theta + \eta \cdot \frac{1}{|\mathcal{B}|} \sum_{i \in \mathcal{B}} (\phi_i - \theta)$     ▷ Meta update, Eq. equation 4
21:     **end for**
22:     **return** $\theta_{\text{REdit}}$
23: **end procedure**

---

## E   Implementation Details

For *circuit reshaping*, we set the inner learning rate to $\alpha = 1 \times 10^{-6}$ and the outer learning rate to $\eta = 1 \times 10^{-6}$, running for 200 steps with an inner update step size of $s = 5$. In each iteration, we sample $|\mathcal{B}| = 2$ contrastive pairs of reasoning patterns in a batch. The temperature for *contrastive circuit shaping* is fixed at $\tau_t = 1$. The *null-space protection* coefficient is set to $\rho = 0.5$, and the *prediction distribution preservation* weight is $\lambda = 0.1$. When computing attribution scores, we use $K = 10$ instantiations for circuit distance calculation and $K = 2$ instantiations for REdit circuit reshaping due to computational restricts. For experiments validating Circuit-Interference Law, we construct circuits with top-$\tau = 5\%$ edges. During editing, we modify one instance per sample. Unless otherwise specified, all editing methods select the optimal learning rate within the range $5 \times 10^{-5}$ and $2 \times 10^{-4}$ for 10 steps. All experiments are conducted on four A100 GPUs; each REdit meta-iteration consumes $\approx$ 1 minute.

**Corrupt Dataset.** To construct the corrupt dataset, we modify the final question to query the status of the first propositional variable in the premise $\mathcal{P}$ (fact-checking), instead of the status of the goal $\mathcal{G}$, while keeping all other components unchanged.

**Prompts.** For the propositional logic dataset, we append the instruction: `(Answer only in True, False, or N/A (Neither)). Answer:` to each question. For the mathematical dataset, we append: `Answer with only the final numeric result. Answer:` to ensure precise and standardized responses.

## F   Case Study

In this section, we present a case study illustrating the circuits of two reasoning patterns before and after REdit circuit reshaping. As shown in Figure 8, prior to reshaping, circuits from different instantiations of reasoning pattern **I** exhibit substantial overlap, though discrepancies remain, most notably around node `a23.h4` and the tree structure formed by `m21`, `m22`, and `m23`. Circuits from reasoning pattern **II** share slight overlap with those of pattern **I**, particularly within the same tree structure. After circuit reshaping, circuits from different instantiations of reasoning pattern **I** become more consistent and exhibit stronger alignment, with noisy

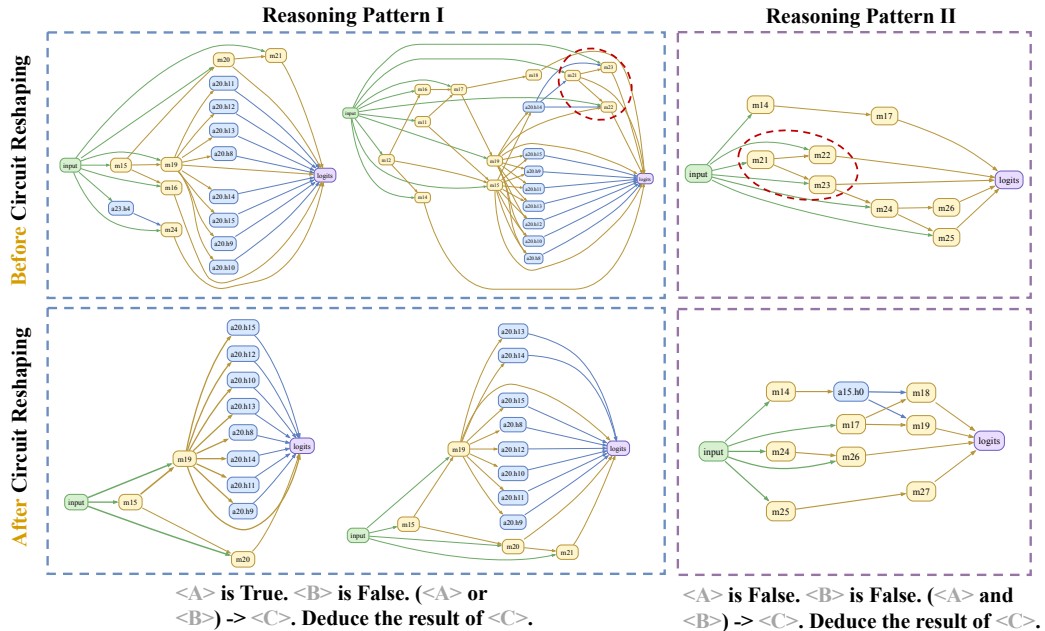

**Reasoning Pattern I**

**Reasoning Pattern II**

<A> is True.  is False. (<A> or
) -> <C>. Deduce the result of <C>.

<A> is False.  is False. (<A> and
) -> <C>. Deduce the result of <C>.

Figure 8: Case study of circuits from reasoning patterns **I** and **II** before and after REdit circuit reshaping. REdit enhances intra-pattern consistency while eliminating inter-pattern overlap.

nodes and edges effectively pruned. At the same time, overlap between circuits of reasoning patterns **I** and **II** is almost completely eliminated. This case study highlights the effectiveness of REdit: it reshapes circuits to achieve greater separation across different reasoning patterns while producing more coherent and centralized structures within the same reasoning pattern.

# G GENERALITY-LOCALITY TRADE-OFF OF REDIT

In this section, we compare the generality–locality trade-off before and after applying circuit reshaping with REdit. As shown in Figure 9, across different learning rates, LLMs trained with REdit consistently achieve a superior Pareto frontier compared to raw LLMs, highlighting the effectiveness of our approach.

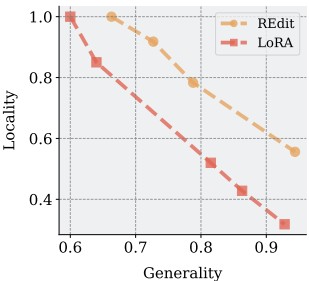

# H ADDITIONAL EXPERIMENTS

Figure 9: Trade-off of REdit

To strengthen empirical support and demonstrate generality beyond
a single architecture and domain, we present two additional sets of experiments.

**Experiments on Gemma-3-1B-IT.** To verify that REdit's effectiveness is not tied to a specific model family, we evaluate all methods using `Gemma-3-1B-IT` as the backbone on ContextHub. As shown in Table 5, REdit consistently outperforms all baselines across all three difficulty levels in generality and achieves competitive locality, confirming that the benefits of circuit reshaping transfer across model architectures. Note that *Raw*'s locality is trivially N/A since no edits have been applied.

**Experiments on the Date Dataset.** To further assess generalization to a qualitatively different reasoning domain, we evaluate REdit on a variant of the commonly used Date Understanding dataset (Srivastava et al., 2023), where each template encodes a temporal reasoning pattern such as computing a date offset from a given reference. Table 6 shows a representative instantiation.

| Dataset | Metric | Raw | BIMT | LoRA | ROME | AlphaEdit | REdit |
|---------|--------|-----|------|------|------|-----------|-------|
| Level 1 | **Generality** | $47.1 \pm 0.1$ | $68.3 \pm 0.1$ | $64.8 \pm 0.4$ | $57.3 \pm 0.1$ | $69.4 \pm 0.4$ | $\mathbf{70.8} \pm 0.1$ |
|         | **Locality** | N/A | $72.2 \pm 0.3$ | $76.3 \pm 0.6$ | $76.5 \pm 0.1$ | $76.3 \pm 2.1$ | $\mathbf{80.6} \pm 0.5$ |
| Level 2 | **Generality** | $46.9 \pm 0.3$ | $60.4 \pm 0.1$ | $55.4 \pm 0.4$ | $56.3 \pm 0.4$ | $56.0 \pm 0.1$ | $\mathbf{69.1} \pm 0.1$ |
|         | **Locality** | N/A | $73.8 \pm 0.4$ | $73.7 \pm 0.1$ | $76.5 \pm 1.1$ | $73.4 \pm 0.1$ | $78.2 \pm 0.1$ |
| Level 3 | **Generality** | $55.3 \pm 0.1$ | $62.6 \pm 0.1$ | $62.0 \pm 0.4$ | $59.5 \pm 0.4$ | $62.3 \pm 0.5$ | $\mathbf{65.6} \pm 0.2$ |
|         | **Locality** | N/A | $87.6 \pm 0.1$ | $87.0 \pm 0.4$ | $\mathbf{88.0} \pm 0.1$ | $87.3 \pm 0.2$ | $87.3 \pm 0.1$ |

Table 5: Results on ContextHub with `Gemma-3-1B-IT` as the backbone. The best scores are highlighted in **bold**.

| Date Template | Instantiated Question |
|---------------|----------------------|
| Today is Christmas Eve of `[YEAR]`. What is the date tomorrow in MM/DD/YYYY? A.`[CHOICE_A]` B.`[CHOICE_B]` C.`[CHOICE_C]` D.`[CHOICE_D]` E.`[CHOICE_E]` F.`[CHOICE_F]` | Today is Christmas Eve of 1909. What is the date tomorrow in MM/DD/YYYY? A. 10/25/1909 B. 12/26/1909 C. 12/28/1909 D. 12/25/1910 E. 12/27/1909 F. 12/25/1909 |

Table 6: Example template and instantiated question from the Date dataset.

As reported in Table 7, REdit achieves the highest generality among all methods and competitive locality on this dataset, outperforming most baselines on both metrics. This further demonstrates the broad applicability of our approach beyond propositional logic and mathematical reasoning.

| Metric | Raw | BIMT | LoRA | ROME | AlphaEdit | REdit |
|--------|-----|------|------|------|-----------|-------|
| **Generality** | $41.5 \pm 1.1$ | $55.3 \pm 1.0$ | $44.2 \pm 0.2$ | $49.5 \pm 0.7$ | $50.8 \pm 0.4$ | $\mathbf{57.6} \pm 0.3$ |
| **Locality** | N/A | $67.8 \pm 0.3$ | $87.6 \pm 0.8$ | $\mathbf{91.2} \pm 1.1$ | $89.2 \pm 1.1$ | $90.7 \pm 0.5$ |

Table 7: Results on the Date dataset with `Qwen-2.5-3B`. The best scores are highlighted in **bold**.

