# OpenReview forum: "Reforming the Mechanism: Editing Reasoning Patterns in LLMs with Circuit Reshaping"
_ICLR.cc/2026/Conference — ICLR 2026 Poster_

### Official Review · Reviewer_cGfW · 2025-10-25

**Soundness:** 3
**Presentation:** 3
**Contribution:** 3
**Rating:** 6
**Confidence:** 4

**Summary:**

This paper introduces REdit, a framework for reasoning editing in LLMs that extends model editing from factual correction to logical reasoning. It proposes the Circuit–Interference Law, linking neural circuit overlap with edit interference, and applies contrastive circuit reshaping with meta-learning and dual-level protection.

Overall, it is well-motivated, technically sound, and creative, addressing how to modify reasoning pathways without interference.

**Strengths:**

The paper presents a novel perspective showing that modifying internal circuit traces can directly influence the final reasoning behavior of LLMs, which is an intriguing and valuable insight.

The proposed Circuit–Interference Law provides a principled and empirically grounded explanation connecting neural circuit overlap with editing interference, representing a fresh and meaningful contribution to mechanistic interpretability research.

**Weaknesses:**

The experimental setting is limited. The authors evaluate only on a single backbone model (Qwen-2.5-3B) and a single dataset, which constrains the generality of the conclusions.

The reported improvements are modest rather than substantial, leaving some uncertainty about the practical effectiveness and scalability of the proposed approach.

**Questions:**

Confused on the instance-specific noise. if edge attribution values vary substantially across samples, how are these aggregated? Is the same top-$\tau$ threshold applied uniformly, or does it adapt to distributional variance across instances?

Could the authors provide specific examples of “reasoning patterns” to clarify how they are defined and represented? Additionally, how do these reasoning patterns change after applying REdit—is there any qualitative or quantitative visualization of this transformation?

---

> ### Author Response · Authors · 2025-11-20
>
> We really appreciate your positive and instructive feedback and will provide piecewise response below.
>
> > **W1:** The experimental setting is limited. The authors evaluate only on a single backbone model (Qwen-2.5-3B) and a single dataset, which constrains the generality of the conclusions.
>
> **R1:** We appreciate the feedback regarding the limited scope of our experimental validation. To strengthen the empirical support and demonstrate generality beyond a single architecture, we have **added experiments on the Gemma-3-1B-IT model**, as reported in the table below (N/A indicates Raw’s locality is trivially 100%). The results show that our method continues to outperform baselines across all levels in ContextHub, indicating that its effectiveness is not tied to a specific model family. Due to resource constraints, we focus our experiments on Qwen-2.5-3B in the paper instead of larger-scale models, noting that REdit stores attribution circuits for multiple samples, which has associated memory requirements. In addition, the reasoning ability of Qwen-2.5-3B is comparable to stronger models (e.g., Llama-7B), so we do not expect qualitatively different behavior at 7B scale [1].
>
> | ContextHub/Gemma-3-1B-IT | Raw | LoRA |BIMT | AlphaEdit | ROME | REdit |
> |----------------------|:----:|:----:|:----:|:----:|:----:|:----:|
> |Level_1_generality|47.1±0.1|64.8±0.4|68.3±0.1|69.4±0.4|57.3±0.1|**70.8±0.1**|
> |Level_1_locality|N/A|76.3±0.6|72.2±0.3|76.3±2.1|76.5±0.1|**80.6±0.5**|
> |Level_2_generality|46.9±0.3|55.4±0.4|60.4±0.1|56.0±0.1|56.3±0.4|**69.1±0.1**|
> |Level_2_locality|N/A|73.7±0.1|73.8±0.4|73.4±0.1|76.5±1.1|**78.2±0.1**|
> |Level_3_generality|55.3±0.1|62.0±0.4|62.6±0.1|62.3±0.5|59.5±0.4|**65.6±0.2**|
> |Level_3_locality|N/A|87.0±0.4|87.6±0.1|87.3±0.2|**88.0±0.1**|87.3±0.1|
>
> In addition to ContextHub, we also evaluate REdit on TemplateGSM (Figure 6), which focuses on mathematical reasoning. To further assess generalization, we **utilize a variant of the commonly used Date dataset** [2], where the underlying reasoning patterns can be loosely defined, with examples shown below and results summarized in the table below. On this dataset as well, REdit consistently outperforms baseline editing methods in both generality and locality, providing a foundation for future extensions to more open-ended, real-world data.
>
> | Date Template| Instantiated Question |
> |----------------------|----|
> |Today is Christmas Eve of **[YEAR]**. What is the date tomorrow in MM/DD/YYYY?\nA. **[CHOICE_A]**\nB. **[CHOICE_B]**\nC. **[CHOICE_C]**\nD. **[CHOICE_D]**\nE. **[CHOICE_E]**\nF. **[CHOICE_F]**|Today is Christmas Eve of 1909. What is the date tomorrow in MM/DD/YYYY?\nA. 10/25/1909\nB. 12/26/1909\nC. 12/28/1909\nD. 12/25/1910\nE. 12/27/1909\nF. 12/25/1909|
>
>
>
> | Date/Qwen-2.5-3B| Raw | LoRA |BIMT | AlphaEdit | ROME | REdit |
> |----------------------|:----:|:----:|:----:|:----:|:----:|:----:|
> |Generality|41.5±1.1|44.2±0.2|55.3±1.0|50.8±0.4|49.5±0.7|**57.6±0.3**|
> |Locality|N/A|87.6±0.8|67.8±0.3|89.2±1.1|**91.2±1.1**|90.7±0.5|
>
>
> > **W2:** The reported improvements are modest rather than substantial, leaving some uncertainty about the practical effectiveness and scalability of the proposed approach.
>
> **R2:** We believe the results are practically meaningful, especially given the difficulty of simultaneously improving generality and perserving locality. REdit achieves **clear gains** over existing methods. For example, on ContextHub Level 1, it improves Generality over the second-best BIMT by 1.9% (from 72.2 to 74.1) and Locality by 32.8% (from 61.5 to 94.3), indicating a much better balance between successful edits and preservation of unrelated behaviors. Across Levels 1–3, REdit further improves average Generality over other model-editing baselines by 6.2%, 4.3%, and 2.9%, respectively, while maintaining competitive or better Locality, demonstrating its practical advantage in real editing scenarios.

---

> ### Author Response · Authors · 2025-11-20
>
> > **Q1:** Confused on the instance-specific noise. if edge attribution values vary substantially across samples, how are these aggregated? Is the same top-threshold applied uniformly, or does it adapt to distributional variance across instances?
>
> **R3:** We follow the **standard convention** in circuit-level analysis of aggregating instance-wise attributions by averaging them over all samples of a given task/pattern [3]. Intuitively, edges that are truly important across instances will remain large under this averaging, while instance-specific noise tends to cancel out. After obtaining the averaged attribution values, we apply the same global top threshold uniformly to select circuit edges.
>
>
> > **Q2:** Could the authors provide specific examples of “reasoning patterns” to clarify how they are defined and represented? Additionally, how do these reasoning patterns change after applying REdit—is there any qualitative or quantitative visualization of this transformation?
>
>
> **R4:** We appreciate the concern about qualitative evidence. In fact, we **do provide a qualitative case study in Appendix F**. There, we visualize the circuits of two reasoning patterns before and after REdit’s circuit reshaping (Figure 8). Before reshaping, circuits from different instantiations of reasoning pattern I show substantial but imperfect overlap, with discrepancies around node a23.h4 and the tree formed by m21, m22, and m23. Circuits for reasoning pattern II partially overlap with those of pattern I, especially within this same subtree. After circuit reshaping, circuits from different instantiations of reasoning pattern I become more consistent and better aligned, with noisy nodes and edges effectively pruned, while the overlap between circuits of patterns I and II is almost entirely removed. This provides qualitative evidence that REdit sharpens and disentangles circuit structure in an interpretable way.
>
>
> [1] Yang, An, et al. "Qwen2.5 Technical Report."
>
> [2] Srivastava, Aarohi, et al. "Beyond the imitation game: Quantifying and extrapolating the capabilities of language models." Transactions on machine learning research (2023).
>
> [3] Syed, Aaquib, Can Rager, and Arthur Conmy. "Attribution patching outperforms automated circuit discovery." Proceedings of the 7th BlackboxNLP Workshop: Analyzing and Interpreting Neural Networks for NLP. 2024.

---

> > ### Comment · Reviewer_cGfW · 2025-11-25
> >
> > Thanks for your response. I believe this is an important step in using interpretation to increase LLMs' performance. I would like to raise my score.

---

> > > ### Author Response · Authors · 2025-11-25
> > >
> > > Thank you so much for the response! We greatly appreciate the time you have taken to read through our replies and raise score. If there are any outstanding issues or further clarifications needed, please do not hesitate to let us know.

---

### Official Review · Reviewer_33G2 · 2025-10-27

**Soundness:** 4
**Presentation:** 3
**Contribution:** 4
**Rating:** 6
**Confidence:** 3

**Summary:**

The paper proposes reasoning editing, which targets edits on reasoning patterns using the proposed REdit method. Through experiments, the authors also uncovered the Circuit-Interference Law, which states that interference between circuits is proportional to their overlap.

**Strengths:**

- The paper's idea is very interesting and shifts attention from factual knowledge toward the logic applied by models, which is a source of many flaws in their performance.
- The Circuit-Interference Law is a significant novelty for the community.

**Weaknesses:**

- The datasets and models used are limited. While it does not seem like the conclusions would differ with larger models, the use of the well-controlled ContextHub raises questions about how things might go wrong with wild, real-world data.

**Questions:**

N/A

---

> ### Author Response · Authors · 2025-11-20
>
> We really appreciate your positive and instructive feedback and will provide piecewise response below.
>
> > **W1:** The datasets and models used are limited. While it does not seem like the conclusions would differ with larger models, the use of the well-controlled ContextHub raises questions about how things might go wrong with wild, real-world data.
>
> **R1:** We appreciate the feedback regarding the limited scope of our experimental validation. To strengthen the empirical support and demonstrate generality beyond a single architecture, we have **added experiments on the Gemma-3-1B-IT model**, as reported in the table below (N/A indicates Raw’s locality is trivially 100%). The results show that our method continues to outperform baselines across all levels in ContextHub, indicating that its effectiveness is not tied to a specific model family. Due to resource constraints, we focus our experiments on Qwen-2.5-3B in the paper instead of larger-scale models, noting that REdit stores attribution circuits for multiple samples, which has associated memory requirements. In addition, the reasoning ability of Qwen-2.5-3B is comparable to stronger models (e.g., Llama-7B), so we do not expect qualitatively different behavior at 7B scale [1].
>
> | ContextHub/Gemma-3-1B-IT | Raw | LoRA |BIMT | AlphaEdit | ROME | REdit |
> |----------------------|:----:|:----:|:----:|:----:|:----:|:----:|
> |Level_1_generality|47.1±0.1|64.8±0.4|68.3±0.1|69.4±0.4|57.3±0.1|**70.8±0.1**|
> |Level_1_locality|N/A|76.3±0.6|72.2±0.3|76.3±2.1|76.5±0.1|**80.6±0.5**|
> |Level_2_generality|46.9±0.3|55.4±0.4|60.4±0.1|56.0±0.1|56.3±0.4|**69.1±0.1**|
> |Level_2_locality|N/A|73.7±0.1|73.8±0.4|73.4±0.1|76.5±1.1|**78.2±0.1**|
> |Level_3_generality|55.3±0.1|62.0±0.4|62.6±0.1|62.3±0.5|59.5±0.4|**65.6±0.2**|
> |Level_3_locality|N/A|87.0±0.4|87.6±0.1|87.3±0.2|**88.0±0.1**|87.3±0.1|
>
> In addition to ContextHub, we also evaluate REdit on TemplateGSM (Figure 6), which focuses on mathematical reasoning. To further assess generalization, we **utilize a variant of the commonly used Date dataset** [2], where the underlying reasoning patterns can be loosely defined, with examples shown below and results summarized in the table below. On this dataset as well, REdit consistently outperforms most baselines in both generality and locality, further demonstrating the robustness and broad applicability of our approach.
>
> | Date Template| Instantiated Question |
> |----------------------|----|
> |Today is Christmas Eve of **[YEAR]**. What is the date tomorrow in MM/DD/YYYY?\nA. **[CHOICE_A]**\nB. **[CHOICE_B]**\nC. **[CHOICE_C]**\nD. **[CHOICE_D]**\nE. **[CHOICE_E]**\nF. **[CHOICE_F]**|Today is Christmas Eve of 1909. What is the date tomorrow in MM/DD/YYYY?\nA. 10/25/1909\nB. 12/26/1909\nC. 12/28/1909\nD. 12/25/1910\nE. 12/27/1909\nF. 12/25/1909|
>
>
>
> | Date/Qwen-2.5-3B| Raw | LoRA |BIMT | AlphaEdit | ROME | REdit |
> |----------------------|:----:|:----:|:----:|:----:|:----:|:----:|
> |Generality|41.5±1.1|44.2±0.2|55.3±1.0|50.8±0.4|49.5±0.7|**57.6±0.3**|
> |Locality|N/A|87.6±0.8|67.8±0.3|89.2±1.1|**91.2±1.1**|90.7±0.5|
>
> [1] Yang, An, et al. "Qwen2.5 Technical Report."
>
> [2] Srivastava, Aarohi, et al. "Beyond the imitation game: Quantifying and extrapolating the capabilities of language models." Transactions on machine learning research (2023).

---

> > ### Author Response · Authors · 2025-11-27
> >
> > Thank you for your time and effort in reviewing our paper! As the deadline for rebuttal is approaching, we would like to kindly ask if you have any remaining questions; we are more than happy to address them.

---

### Official Review · Reviewer_Qw78 · 2025-10-29

**Soundness:** 4
**Presentation:** 4
**Contribution:** 3
**Rating:** 6
**Confidence:** 3

**Summary:**

The article presents the reasoning editing method for point-by-point modification of logical reasoning patterns in language models without losing other skills. The authors strike a balance between generalization and locality, describe the law of circuit interference, and propose the REdit system. Tests on logical and mathematical tasks show that REdit outperforms existing methods, paving the way for more precise control over model reasoning.

**Strengths:**

* The shift from knowledge editing to reasoning editing is original and well-motivated. The generality–locality trade-off is crisply formulated and backed by empirical evidence
* The results include confidence intervals, which make them more reliable and indicate the stability of the findings.

**Weaknesses:**

* The experiments focus only on propositional logic and structured math tasks, so they don’t fully reflect how reasoning works in more open-ended, real-world settings. It’s still unclear whether the method would hold up beyond these controlled, symbolic cases.
* While “circuits” are central, empirical evidence that reshaped circuits correspond to interpretable submodules is limited to correlation plots. There’s no qualitative analysis of what circuits actually represent.
* The approach is evaluated on a single model Qwen-2.5-3B, so it’s unclear whether the results would hold across different architectures or model scales.

**Questions:**

See weaknesses

---

> ### Author Response · Authors · 2025-11-20
>
> We really appreciate your positive and instructive feedback and will provide piecewise response below.
>
> > **W1:** The experiments focus only on propositional logic and structured math tasks, so they don’t fully reflect how reasoning works in more open-ended, real-world settings. It’s still unclear whether the method would hold up beyond these controlled, symbolic cases.
>
> **R1:** We appreciate the concern about generalizing to more real-world reasoning. At present, our method assumes that the required reasoning pattern for each question is known in advance, which is explicitly provided by datasets like ContextHub and TemplateMath, but not by more open-ended benchmarks.
>
> To extend beyond these constrained datasets, we have **added experiments on a variant of the commonly used Date dataset** [1], where the underlying reasoning patterns can be loosely defined. We observe that REdit continues to perform effectively in this setting as well, suggesting that its benefits extend beyond explicitly structured symbolic tasks.
>
> | Date Template| Instantiated Question |
> |----------------------|----|
> |Today is Christmas Eve of **[YEAR]**. What is the date tomorrow in MM/DD/YYYY?\nA. **[CHOICE_A]**\nB. **[CHOICE_B]**\nC. **[CHOICE_C]**\nD. **[CHOICE_D]**\nE. **[CHOICE_E]**\nF. **[CHOICE_F]**|Today is Christmas Eve of 1909. What is the date tomorrow in MM/DD/YYYY?\nA. 10/25/1909\nB. 12/26/1909\nC. 12/28/1909\nD. 12/25/1910\nE. 12/27/1909\nF. 12/25/1909|
>
>
>
> | Date/Qwen-2.5-3B| Raw | LoRA |BIMT | AlphaEdit | ROME | REdit |
> |----------------------|:----:|:----:|:----:|:----:|:----:|:----:|
> |Generality|41.5±1.1|44.2±0.2|55.3±1.0|50.8±0.4|49.5±0.7|**57.6±0.3**|
> |Locality|N/A|87.6±0.8|67.8±0.3|89.2±1.1|**91.2±1.1**|90.7±0.5|
>
> More broadly, we see automatic discovery of the underlying reasoning patterns for open-ended questions as an important and challenging direction. Extending REdit to such scenarios where patterns must be inferred rather than given is a natural next step and a focus of our future work.
>
>
> > **W2:** While “circuits” are central, empirical evidence that reshaped circuits correspond to interpretable submodules is limited to correlation plots. There’s no qualitative analysis of what circuits actually represent.
>
> **R2:** We appreciate the concern about qualitative evidence. In fact, we **do provide a qualitative case study in Appendix F**. There, we visualize the circuits of two reasoning patterns before and after REdit’s circuit reshaping (Figure 8). Before reshaping, circuits from different instantiations of reasoning pattern I show substantial but imperfect overlap, with discrepancies around node a23.h4 and the tree formed by m21, m22, and m23. Circuits for reasoning pattern II partially overlap with those of pattern I, especially within this same subtree. After circuit reshaping, circuits from different instantiations of reasoning pattern I become more consistent and better aligned, with noisy nodes and edges effectively pruned, while the overlap between circuits of patterns I and II is almost entirely removed. This provides qualitative evidence that REdit sharpens and disentangles circuit structure in an interpretable way.
>
> Going further toward fully interpretable submodules for propositional logic in ContextHub would require substantial, manual interpretability analysis and dedicated methodology, which we view as important but orthogonal future work, potentially worthy of a separate paper.

---

> ### Author Response · Authors · 2025-11-20
>
> > **W3:** The approach is evaluated on a single model Qwen-2.5-3B, so it’s unclear whether the results would hold across different architectures or model scales.
>
> **R3:** To strengthen the empirical support and assess generality across architectures and model scales, we have **added experiments on Gemma-3-1B-IT**, as reported in the table below (N/A indicates Raw’s locality is trivially 100%). The results show that REdit consistently outperforms baselines across all levels on ContextHub, indicating that its effectiveness is not tied to a specific model family. Due to resource constraints, we focus our experiments on Qwen-2.5-3B in the paper instead of larger-scale models, noting that REdit stores attribution circuits for multiple samples, which has associated memory requirements. In addition, the reasoning ability of Qwen-2.5-3B is comparable to stronger models (e.g., Llama-7B), so we do not expect qualitatively different behavior at 7B scale [2].
>
>
> | ContextHub/Gemma-3-1B-IT | Raw | LoRA |BIMT | AlphaEdit | ROME | REdit |
> |----------------------|:----:|:----:|:----:|:----:|:----:|:----:|
> |Level_1_generality|47.1±0.1|64.8±0.4|68.3±0.1|69.4±0.4|57.3±0.1|**70.8±0.1**|
> |Level_1_locality|N/A|76.3±0.6|72.2±0.3|76.3±2.1|76.5±0.1|**80.6±0.5**|
> |Level_2_generality|46.9±0.3|55.4±0.4|60.4±0.1|56.0±0.1|56.3±0.4|**69.1±0.1**|
> |Level_2_locality|N/A|73.7±0.1|73.8±0.4|73.4±0.1|76.5±1.1|**78.2±0.1**|
> |Level_3_generality|55.3±0.1|62.0±0.4|62.6±0.1|62.3±0.5|59.5±0.4|**65.6±0.2**|
> |Level_3_locality|N/A|87.0±0.4|87.6±0.1|87.3±0.2|**88.0±0.1**|87.3±0.1|
>
> [1] Srivastava, Aarohi, et al. "Beyond the imitation game: Quantifying and extrapolating the capabilities of language models." Transactions on machine learning research (2023).
>
> [2] Yang, An, et al. "Qwen2.5 Technical Report."

---

> ### Comment · Reviewer_Qw78 · 2025-11-25
>
> Thank you for the clear and helpful responses. The additional experiments address my concerns, and I am raising my score.

---

> > ### Author Response · Authors · 2025-11-26
> >
> > We sincerely thank you for recognition of our efforts and raise score. If there are any outstanding issues or further clarifications needed, please do not hesitate to let us know.

---

### Official Review · Reviewer_ro7e · 2025-10-29

**Soundness:** 3
**Presentation:** 3
**Contribution:** 3
**Rating:** 6
**Confidence:** 4

**Summary:**

The author proposed REdit, which is one of the pioneering works in reasoning editing. The work gives a perspective from the knowledge circuit on the reasoning process, which is inspiring and interesting. Specifically, the author proposed a systematic framework for reasoning editing with the emphasis on neuron circuits, and the Redit framework combines the findings together to achieve the editing.

**Strengths:**

- The presentation of the paper is well-organized and convincing

- The perspective of the knowledge circuit and the finding of Circuit-Interference Law is inspiring and novel.

**Weaknesses:**

- The work lacks sufficient experimental validation. The current experiments in Sec. 4 are conducted on only one model (Qwen-2.5-3B) and one dataset (ContextHub), which limits the generality of the conclusions.

- The choice of base model is inconsistent between the preliminary analysis in Sec. 2.2 and the main experiments in Sec. 4. In addition, the experimental setup in Sec. 3.1 is not clearly described (not sure if this is also based on Qwen-2.5-3B).

- The method appears overly complex, but the final results do not show clear improvement, leaving me unconvinced about its effectiveness and practical value.

Overall, I am very interested in the ideas and perspectives in this work. It would be helpful for enhancing this paper if the author can provide more results on more base models and datasets to show the effectiveness of Redit.

**Questions:**

See above

---

> ### Author Response · Authors · 2025-11-20
>
> We really appreciate your positive and instructive feedback and will provide piecewise response below.
>
> > **W1:** The work lacks sufficient experimental validation. The current experiments in Sec. 4 are conducted on only one model (Qwen-2.5-3B) and one dataset (ContextHub), which limits the generality of the conclusions.
>
> **R1:** We appreciate the feedback regarding the limited scope of our experimental validation. To strengthen the empirical support and demonstrate generality beyond a single architecture, we have **added experiments on the Gemma-3-1B-IT model**, as reported in the table below (N/A indicates Raw’s locality is trivially 100%). The results show that our method continues to outperform baselines across all levels in ContextHub, indicating that its effectiveness is not tied to a specific model family.
>
> | ContextHub/Gemma-3-1B-IT | Raw | LoRA |BIMT | AlphaEdit | ROME | REdit |
> |----------------------|:----:|:----:|:----:|:----:|:----:|:----:|
> |Level_1_generality|47.1±0.1|64.8±0.4|68.3±0.1|69.4±0.4|57.3±0.1|**70.8±0.1**|
> |Level_1_locality|N/A|76.3±0.6|72.2±0.3|76.3±2.1|76.5±0.1|**80.6±0.5**|
> |Level_2_generality|46.9±0.3|55.4±0.4|60.4±0.1|56.0±0.1|56.3±0.4|**69.1±0.1**|
> |Level_2_locality|N/A|73.7±0.1|73.8±0.4|73.4±0.1|76.5±1.1|**78.2±0.1**|
> |Level_3_generality|55.3±0.1|62.0±0.4|62.6±0.1|62.3±0.5|59.5±0.4|**65.6±0.2**|
> |Level_3_locality|N/A|87.0±0.4|87.6±0.1|87.3±0.2|**88.0±0.1**|87.3±0.1|
>
>
> In addition to ContextHub, we also **evaluate REdit on TemplateGSM** (Figure 6), which focuses on mathematical reasoning. To further assess generalization, we **utilize a variant of the commonly used Date dataset** [1], where the underlying reasoning patterns can be loosely defined, with examples shown below and results summarized in the table below. On this dataset as well, REdit consistently outperforms most baselines in both generality and locality, further demonstrating the robustness and broad applicability of our approach.
>
> | Date Template| Instantiated Question |
> |----------------------|----|
> |Today is Christmas Eve of **[YEAR]**. What is the date tomorrow in MM/DD/YYYY?\nA. **[CHOICE_A]**\nB. **[CHOICE_B]**\nC. **[CHOICE_C]**\nD. **[CHOICE_D]**\nE. **[CHOICE_E]**\nF. **[CHOICE_F]**|Today is Christmas Eve of 1909. What is the date tomorrow in MM/DD/YYYY?\nA. 10/25/1909\nB. 12/26/1909\nC. 12/28/1909\nD. 12/25/1910\nE. 12/27/1909\nF. 12/25/1909|
>
>
>
> | Date/Qwen-2.5-3B| Raw | LoRA |BIMT | AlphaEdit | ROME | REdit |
> |----------------------|:----:|:----:|:----:|:----:|:----:|:----:|
> |Generality|41.5±1.1|44.2±0.2|55.3±1.0|50.8±0.4|49.5±0.7|**57.6±0.3**|
> |Locality|N/A|87.6±0.8|67.8±0.3|89.2±1.1|**91.2±1.1**|90.7±0.5|
>
> > **W2:** The choice of base model is inconsistent between the preliminary analysis in Sec. 2.2 and the main experiments in Sec. 4. In addition, the experimental setup in Sec. 3.1 is not clearly described (not sure if this is also based on Qwen-2.5-3B).
>
> **R2:** Thanks for pointing out the confusion regarding the choice of base models. Due to resource constraints, we focus our experiments on Qwen-2.5-3B in the paper instead of larger-scale models, noting that REdit stores attribution circuits for multiple samples, which has associated memory requirements. In addition, we chose Qwen-2.5-3B as our single base model rather than other small models, because its reasoning ability is comparable to stronger models (e.g., Llama-7B), making it a representative choice for which we do not expect qualitatively different behavior at 7B scale [2]. We also apologize for the lack of clarity in Sec. 3.1. This section is indeed based on Qwen-2.5-3B, and we will revise the text to explicitly state the base model used in this setup.

---

> ### Author Response · Authors · 2025-11-20
>
> > **W3:** The method appears overly complex, but the final results do not show clear improvement, leaving me unconvinced about its effectiveness and practical value.
>
> **R3:** We appreciate the concern about the complexity of our method. The **core idea of REdit is actually simple and intuitive**: we apply contrastive learning on attribution circuits, using positive samples from the same pattern and negative samples from different patterns to disentangle overlapping circuits. This core component alone already yields strong editing performance.
>
> The additional components meta-contrastive learning and dual-level protection are introduced to further improve transferability and robustness, and to better preserve the original model behavior, rather than to add unnecessary complexity.
>
> In terms of effectiveness, REdit **achieves clear gains** over existing methods. For example, on ContextHub Level 1, it improves Generality over the second-best BIMT by 1.9% (from 72.2 to 74.1) and Locality by 32.8% (from 61.5 to 94.3), indicating a much better balance between successful edits and preservation of unrelated behaviors. Across Levels 1–3, REdit further improves average Generality over other model-editing baselines by 6.2%, 4.3%, and 2.9%, respectively, while maintaining competitive or better Locality, demonstrating its practical advantage in real editing scenarios.
>
> [1] Srivastava, Aarohi, et al. "Beyond the imitation game: Quantifying and extrapolating the capabilities of language models." Transactions on machine learning research (2023).
>
> [2] Yang, An, et al. "Qwen2.5 Technical Report."

---

> > ### Comment · Reviewer_ro7e · 2025-11-25
> >
> > Thank you for your response and for providing the additional experiments. I believe these clarifications and results strengthen the paper. My concern about compatibility with existing methods remains, as it would be valuable to demonstrate that the proposed approach can be easily integrated to improve current techniques. Nonetheless, I will maintain my positive score.

---

> ### Author Response · Authors · 2025-11-26
>
> We sincerely thank you for your continued engagement and for recognizing the improvements provided by the additional experiments and clarifications. We also appreciate your constructive suggestion regarding compatibility with existing methods.
>
> We believe that REdit is in fact compatible with existing methods in several ways:
> 1. **Conceptual simplicity and modularity.** The core idea of REdit is contrastive learning on attribution circuits, which can be easily applied to disentangle overlapping circuits and achieve the desired editing effect. This step is modular and can be plugged into various post-pretraining stages.
> 2. **Compatibility across architectures.** Our experiments on multiple modern LLM architectures (e.g., Qwen-2.5-3B and Gemma-3-1B-IT) show that REdit’s mechanism is not tied to a specific model family. This architectural robustness supports its compatibility with methods.
> 3. **Practical complement to scaling-based reasoning improvements.** REdit can be directly utilized to improve LLMs’ logical reasoning abilities without relying on common test-time scaling strategies (which increase inference latency) or wide pretraining (which is training-intensive). At the same time, the dual-level protection design aims to avoid interfering with the models’ existing capabilities.
>
> We hope this addresses your concern, and we are grateful for your thoughtful feedback.

---

### Author Response · Authors · 2025-12-01
**Rebuttal Summary**

Dear Area Chair:

We sincerely thank you for taking on the additional workload caused by the recent OpenReview incident, and we greatly appreciate your effort in re-evaluating our submission under these unusual circumstances. To make your task easier, we briefly summarize the state of the discussion before reviews were rolled back.

### **Key Takeaways Before Rebuttal**

Before the rebuttal period, all four reviewers found our paper **interesting and well-motivated**, assigning **overall positive scores**.

### **Rebuttal Revisions**

In response to the few concerns raised, we provided several key revisions:

1.  Added experiments on an additional architecture (**Gemma-3-1B-IT**).
2.  Clarified evaluations on **TemplateGSM** and included further evaluations on the **Date dataset**.
3.  Clarified the **core mechanism** and the **qualitative circuit analyses**.

### **Reviewer Reactions to Revisions**

After we posted our rebuttal on 20 Nov, reviewers reacted **very positively** to these revisions, and all score changes (yielding an average score of **7.5** [top 0.3%]) occurred before the leakage date (27 Nov):

* **Reviewer Qw78** raised their score from **6 → 8** (25 Nov), stating clearly that the new experiments addressed their concerns.
* **Reviewer cGfW** raised their score from **6 → 10** (24 Nov), describing our work as **“an important step in using interpretation to increase LLMs’ performance.”**
* **Reviewer ro7e** confirmed that our clarifications and additional results **strengthened the paper** and maintained the positive score of **6**, while proposing minor new concerns. We believe our latest response will resolve their new concerns.
* **Reviewer 33G2** provided a positive initial review (score **6**) and has not replied to our rebuttal yet. We believe our new response will resolve their concerns.

### **Reference Links**

We have also provided anonymous snapshot links documenting this pre-rollback state for your reference (before the exploitation of the leakage bug):

* **OpenReview Snapshot:** `https://anonymous.4open.science/r/REdit-DBD8/Rebuttal_State.pdf` (need to download)
* **Paper Copilot Record:** `https://github.com/jingyangcarl/openreview/blob/main/venues/iclr/iclr2026/iclr2026.11262025.12.json`

We hope this summary conveys both our good-faith effort during the rebuttal and the strong recognition from reviewers, and we trust your fair assessment of our work in light of this context.

---

> ### Author Response · Authors · 2025-12-01
> **Recognized Strengths and Contributions**
>
> The reviewers reached a strong consensus regarding the novelty, mechanistic grounding, and empirical robustness of this work:
>
> * **Pioneering Problem Formulation:** Reviewers consistently praised the shift from factual knowledge editing to Reasoning Editing, describing the perspective as "original," "well-motivated," and "inspiring" (Reviewers ro7e, Qw78). They also recognized this as **"paving the way for more precise control"** (Reviewer Qw78) over model reasoning pathways.
> * **Mechanistic Insight:** The proposed Circuit-Interference Law was identified as a major contribution to the interpretability field. Reviewers noted it provides a "principled and empirically grounded explanation" (Reviewer cGfW) and a **"significant novelty"** (Reviewer 33G2) by structurally linking neural circuit overlap to the interference observed during model editing.
> * **Robust Generalization:** The rebuttal experiments introducing the **Gemma-3-1B-IT** architecture and the **Date dataset** successfully demonstrated that REdit is not tied to a specific model family or task. This broader validation directly precipitated score increases from Reviewers Qw78 ($6 \rightarrow 8$) and cGfW ($6 \rightarrow 10$).
>
> In addition to these recognized strengths, we emphasize two additional contributions inherent to our approach:
>
> * **A Practical Complement to Scaling-Based Methods:** REdit offers a practical approach to enhancing LLMs' reasoning capabilities without incurring huge costs associated with test-time scaling strategies (which increase inference latency) or large-scale pretraining (which is computationally intensive).
> * **Active Circuit Reshaping Methodology:** Our work introduces a methodology for active neural circuit modulation in LLMs. This approach moves beyond passive circuit analysis to enable the principled and targeted modification of specific reasoning pathways, achieving intentional and functional circuit reshaping.

---

### Meta-Review · Area_Chair_tLYD · 2025-12-24

**Summary:**

The paper introduces a paradigm for editing reasoning patterns in LLMs. This is in contrast to the majority of the literature on editing, which focuses on fact editing. The authors claim that these methods that are designed for fact editing don’t work well with editing logical reasoning.

The authors also make a claim that the interference of two reasoning patterns during editing is proportional to the overlap of their underlying neural circuits. They then propose a solution that first reshapes the neural circuits (this involves several steps, such as improving disentanglement of these circuits), followed by a lightweight parameter-efficient model editing.

**Reviewer Concerns:**

Reviewers brought up limited experimental scope and task diversity as drawbacks in the initial submission. The authors addressed this by adding another dataset and experiments with Gemma-3-1B-IT.

The reviewers also pointed out that the method seemed overly complex relative to modest gains, raising questions about practical value. Lack of qualitative analysis was also noted, but the authors already had some experiments in the appendix to address this.

Finally, one of the reviewers noted that it is unclear whether the proposed method can be easily combined with existing editing approaches. The authors provided some reasoning why they think that REdit should work with other editing approaches, but I believe no concrete and extensive experiments were provided to demonstrate that this is indeed the case. I would say this is one of the biggest current limitations.

**Reviewer Scores:**

Reviewer ro7e: responded to the rebuttal, mostly satisfied with the additional experiments, but pointed out that the compatibility issues (that i also mention above) has not really been addressed. Overall, I believe this reviewer would have proposed to accept the work.

Reviewer Qw78, Reviewer 33G2, and Reviewer cGfW: their primary concern on adding extra dataset and model has been addressed, therefore, I believe these Reviewers would have recommended an accept.

---

### Decision · Program_Chairs · 2026-01-26

Accept (Poster)